# Assessment of Bioprotect’s Biodegradable Balloon System as a Rectal Spacer in Radiotherapy: An Animal Study on Tissue Response and Biocompatibility

**DOI:** 10.3390/pharmaceutics15122744

**Published:** 2023-12-07

**Authors:** Yuval Ramot, Tal Levin-Harrus, Adva Ezratty, Michal Steiner, Nati Ezov, Abraham J. Domb, Muhammad Abdel-Haq, Shaul Shohat, Liron Aperman, Lee Adler, Oleg Dolkart, Abraham Nyska

**Affiliations:** 1Department of Dermatology, Hadassah Medical Center, Jerusalem 9112001, Israel; yuval.ramot@mail.huji.ac.il; 2Faculty of Medicine, Hebrew University of Jerusalem, Jerusalem 9112001, Israel; 3Envigo CRS Israel Limited, Ness Ziona 7414001, Israel; tal.levinharrus@hbi-cro.com (T.L.-H.); adva.ezratty@hbi-cro.com (A.E.); michal.steiner@hbi-cro.com (M.S.); nati.ezov@hbi-cro.com (N.E.); 4Institute for Drug Research, School of Pharmacy, Faculty of Medicine, Hebrew University of Jerusalem, Jerusalem 9112001, Israel; avid@ekmd.huji.ac.il (A.J.D.); muhammad.abdel-haq@mail.huji.ac.il (M.A.-H.); 5BioProtect, Tzur Yigal 4486200, Israel; shaul@bioprotect.com (S.S.); liron@bioprotect.com (L.A.); lee@bioprotect.com (L.A.); 6Assuta Ashdod University Hospital, Ben-Gurion University of the Negev, Beer Sheba 8410501, Israel; olegdo@assuta.co.il; 7Sackler School of Medicine, Tel Aviv University, Tel Aviv 6200515, Israel

**Keywords:** prostate cancer, rectal spacer, Bioprotect balloon, radiotherapy, biodegradable balloon, tissue response, animal model, radiation therapy, treatment complications, biocompitability

## Abstract

Prostate cancer is a significant health concern for men, emphasizing the need for effective treatment strategies. Dose-escalated external beam radiotherapy shows promise in improving outcomes but presents challenges due to radiation effects on nearby structures, such as the rectum. Innovative techniques, including rectal spacers, have emerged to mitigate these effects. This study comprehensively assessed tissue responses following the implantation of the Bioprotect biodegradable fillable balloon as a rectal spacer in a rat model. Evaluation occurred at multiple time points (4, 26, and 52 weeks) post-implantation. Results revealed localized tissue responses consistent with the expected reaction to biodegradable materials, characterized by mild to moderate fibrotic reactions and encapsulation, underscoring the safety and biocompatibility of the balloon. Importantly, no other adverse events occurred, and the animals remained healthy throughout the study. These findings support its potential clinical utility in radiotherapy treatments to enhance patient outcomes and minimize long-term implant-related complications, serving as a benchmark for future similar studies and offering valuable insights for researchers in the field. In conclusion, the findings from this study highlight the safety, biocompatibility, and potential clinical applicability of the Bioprotect biodegradable fillable balloon as a promising rectal spacer in mitigating radiation-induced complications during prostate cancer radiotherapy.

## 1. Introduction

Prostate cancer ranks as the second-leading cause of mortality in American men following a cancer diagnosis [1]. Remarkably, one in every nine men is anticipated to receive a prostate cancer diagnosis during their lifetime. Consequently, the development of effective treatment regimens for prostate cancer management assumes paramount significance [1]. Dose-escalated external beam radiotherapy (RT) stands out as a compelling strategy for achieving favorable biochemical and clinical outcomes in prostate cancer management [2,3,4,5,6,7,8]. Nonetheless, the anatomical proximity of the prostate to the urinary tract and rectum necessitates consideration, as these adjacent structures are susceptible to the deleterious effects of ionizing radiation. Consequently, while dose escalation is associated with an improved relapse-free survival rate, it also brings about a heightened incidence of adverse effects in the urinary tract and rectum, along with potential complications related to erectile dysfunction [5].

In pursuit of mitigating these undesirable consequences, several innovative technical advancements have emerged. These include the development of intensity-modulated RT (IMRT) [4], the utilization of volume-modulated arc therapy [9,10], and the integration of image-guided RT (IGRT) into clinical practice [7]. Despite the significant strides made in refining these techniques, one challenge endures—the delivery of radiation dose to the rectum, whether through external beam RT or brachytherapy, remains a limiting factor when contemplating dose escalation in prostate cancer treatment. 

The occurrence and intensity of rectal adverse events (AEs) can be mitigated by augmenting the separation between the prostate and the rectum. This is achieved through the insertion or injection of spacers composed of either biodegradable materials, such as hyaluronic acid (HA), or non-biodegradable materials, into the perirectal fat. The underlying principle of using a spacer is to introduce a non-biotoxic substance between the posterior surface of the prostate and the anterior surface of the rectum. This strategic placement effectively reduces the volume of the rectum exposed to high radiation doses [1,11]. Notably, the adoption of spacer techniques has been linked to a reduction in the frequency and severity of rectal AEs [5,12,13,14]. 

The Bioprotect biodegradable fillable balloon is made of PLCL (poly (L-lactide-co-ε-caprolactone) (BioProtect Ltd., Tzur Yigal, Israel). It serves as a rectal spacer that has demonstrated both safety and efficacy in comprehensive preclinical and clinical investigations [5,15,16,17,18,19]. After positioning the balloon in the intended location, it is filled with sterile saline until it achieves its ultimate configuration. The balloon maintains this inflated state throughout the entirety of the treatment duration [16,17]. The formation of the balloons as well as their degradation process have been described in our previous publications [20,21]. 

The utilization of biodegradable materials is progressively gaining traction across various medical disciplines [22]. Therefore, it is imperative that biodegradable materials undergo thorough evaluation through preclinical studies, emphasizing the importance of comprehending the anticipated clinical and pathological outcomes stemming from such investigations. While there is a growing body of knowledge regarding the anticipated pathological findings associated with the use of biodegradable materials in suitable animal models [22], our understanding of biodegradable materials for rectal spacers remains relatively limited. 

The objective of this investigation was to comprehensively evaluate the localized tissue response subsequent to the implantation of a perirectally positioned balloon in a rat model followed by a course of irradiation consisting of five consecutive days to mimic the clinical scenario expected in humans.

## 2. Materials and Methods

### 2.1. Study Design

The study was conducted in Envigo CRS, Israel, following approval by the National Council for Animal Experimentation (Approval No. NPC-En-IL-2111-129-4). Six-week-old male Sprague Dawley [Hsd: Sprague Dawley^®^ SD^®^] rats were procured from Envigo RMS (Ness-Ziona, Israel) LTD. In this study, 30 animals were implanted with the balloon and categorized into a group denoted as Group 2M (Table 1). This group was further subdivided into three sub-groups, each assigned to specific predetermined observation time points (4 weeks, 26 weeks, and 52 weeks), with each sub-group comprising 10 animals. Additionally, an equally sized control group (and corresponding sub-groups) underwent sham operations and were subjected to identical experimental conditions, constituting Group 1M. At the conclusion of the designated observation time points, the animals underwent urinalysis, blood sampling, and were subsequently euthanized. Selected organs were harvested, fixed, and submitted for processing and histopathological evaluation.

### 2.2. Surgical Procedure and Irradiation

Delicate surgical scissors were employed to create a horizontal skin incision approximately 2 cm in length, positioned directly below the anus. Blunt dissection was carefully performed with the scissors, separating the perineal muscles in a ventral direction. This dissection aimed to facilitate visualization of the rectal wall and create a deep pocket beneath it. Particular attention was given to ensuring that the pocket was both sufficiently deep and spacious enough to accommodate the entire implant, with unaffected subcutaneous tissue above it. This design allowed for closure without any contact with the implant and minimized the risk of implant extrusion through the skin incision. The balloon was then inserted into the created pocket, adjacent to the rectal wall and positioned as deeply as possible. To secure the balloon in place, it was affixed to the subcutaneous tissue on each lateral side of the implant, utilizing 5-0 non-absorbable suture material (Polypropylene monofilament) with a simple interrupted suture applied on each side. The skin incision was closed using surgical wound clips.

In the sham-operated group, the animals underwent an identical procedure to that described for the balloon-implanted groups. This included the placement of non-absorbable sutures on both lateral sides of the created pocket. However, it is important to note that in the sham-operated group, no implant material was inserted into the pocket.

Each animal underwent a course of irradiation consisting of five consecutive days, commencing seven days after implantation. The irradiation was administered using the Poskom pxr 1 tube series x-ray generator. To deliver a dose of 2 Gray per fraction, the following parameters were employed: four consecutive pulses were generated for each animal, with the animal positioned approximately 0.5 cm from the collimator. The machine settings were configured to 61 kV, 160 mA, and 2 s (320 mAs). The collimator was adjusted to yield an exposure field measuring 0.5–1.0 cm^2^, and it was expected that each individual pulse would produce a dose of 0.5 Gray.

### 2.3. In Life Analysis

The animals underwent observation for a cumulative period spanning 4 weeks, 26 weeks, and 52 weeks post-implantation. Viability assessments, which included monitoring for mortality and overall condition, were conducted once daily throughout the entire 52-week observation period. Additionally, Cage-Side Clinical Observations were performed once daily over the same 52-week period. Comprehensive clinical examinations, encompassing both systemic and local reactions, were conducted on all animals one day after implantation and subsequently once weekly. The determination of individual body weights of animals was initially conducted during the randomization procedure, followed by body weight assessments on the day of implantation and subsequently on a weekly basis. Individual urine samples were collected during the last week prior to each scheduled termination for the animals. 

### 2.4. Terminal Investigations

At designated time points, animals were euthanized via CO_2_ asphyxiation. Prior to each termination, hematology and biochemistry parameters were assessed. A comprehensive necropsy, including gross pathological examination of the implantation sites, was conducted on all animals. Specific attention was paid to macroscopic alterations, regional lymph nodes, tissue reactions (such as hematoma, edema, encapsulation), and implant characteristics. For the implantation site near the rectum, adjacent tissues including the anus and surrounding unaffected tissue (2 mm to 5 mm) were excised for histopathological evaluation. In cases where the implant was not visible, additional tissue surrounding the expected implant site was included. Collection of regional lymph nodes, kidneys, and any abnormalities was also performed during necropsy. 

### 2.5. Histopathological Assessment

The following organs/tissues underwent processing and evaluation: the implantation site, regional draining lymph nodes, and kidneys. Tissues were meticulously trimmed, embedded in paraffin, and sectioned to approximately 5-micron thickness. Subsequent staining with Hematoxylin and Eosin (H&E) was performed. Special care was taken to maintain the integrity of the implant/tissue interface during the staining process for the implantation site. Xylene-free reagents were used for deparaffinization to prevent implant displacement.

For the implantation site, three transverse sections were prepared, covering the entire site and adjacent rectum. Each transverse slice was embedded in paraffin to expose the cut surface, resulting in one block per slice and three slides per animal for examination.

Histopathological examination of the implantation site involved assessing various parameters, including necrosis graded on a scale from 0 (none) to 4 (severe). Additionally, evaluations were conducted for neovascularization, fibrosis/fibrous capsule formation, and fatty infiltrate, with scoring according to a specific evaluation system detailed in Table 2. 

The histopathological evaluation involved assessing the quantity and dispersion of different inflammatory cell types concerning their proximity to the implantation site/tissue interface. These cell types included polymorphonuclear neutrophilic leukocytes, lymphocytes, plasma cells, eosinophils, macrophages, and multinucleated cells. The assessment of cell types and their responses utilized the evaluation system detailed in Table 3. 

We calculated the total scores for each parameter in both treatment and control implantation sites. Additionally, we doubled the scores for inflammatory cell infiltrates and necrosis. These combined scores were then averaged for each group. To establish a reactivity grade, we subtracted the average score of the control treatment from that of the test treatment, utilizing the interpretation scale detailed in Table 4.

At the implantation site, specific parameters were assessed and graded on a severity scale, including:Incidents of hemorrhage;Granuloma formation;Presence of fragmentation and/or debris;Presence and location of degraded material remnants;Quantity and quality of tissue ingrowth;Mineralization.

These parameters were graded on a severity scale ranging from 0 (none) to 4 (severe). Additionally, the quantity of residual implanted material in balloon-implanted sites was evaluated and scored using a specific grading scale [23]:

0 = No apparent degradation.

1 = Minimal degradation of implant, with some minor dissolution on edges, cracks in the implant, and/or small fragments present.

2 = Moderate degradation of implant with cracks in the implant and/or some fragments.

3 = Marked degradation of implant with the presence of several fragments.

4 = Abundant degradation of implant with (almost) complete fragmentation.

Any additional histopathological changes or changes observed in other organs were described and scored. These assessments utilized a semi-quantitative grading system with five grades (0–4), taking into consideration the severity of the changes [24]: Grade 0 = no changes observed; Grade 1 = minimal; Grade 2 = mild; Grade 3 = moderate; Grade 4 = severe.

### 2.6. Statistical Analysis

#### 2.6.1. Calculations

MeanSDRelative_01.2.Rnw: this is a validated R-Script used for calculations of mean group, standard deviation, and the number of observations.MultiComp.Rnw: this is a validated R-Script used for statistical evaluations involving multiple groups and/or multiple parameters between two groups.

#### 2.6.2. Evaluation Process

Before applying the appropriate statistical method, a normality test was performed to determine if the data followed a Gaussian distribution, for example, using the Shapiro–Wilk normality test with a significance level of *p* < 0.01.

For MultiComp.Rnw, the following steps were taken based on the normality test results:If the normality test passed for all groups:
An equal-variance test (e.g., Bartlett test) was performed with a significance level of *p* < 0.01.If the Bartlett test passed, a one-way ANOVA with Dunnett’s posttest was performed.If the Bartlett test did not pass, a Kruskal–Wallis test with Mann–Whitney U test was performed.
If the normality test did not pass for all groups, a Kruskal–Wallis test with Mann–Whitney U test was performed for further analysis.

## 3. Results 

### 3.1. Mortality and Clinical Signs

No mortality occurred in any of the animals evaluated during the entire observation period. Throughout the 52-week observation period, no abnormal clinical signs were observed, except for two transient incidents:One incidence of alopecia on the scrotum was noted in a balloon-implanted animal during the second week post-implantation.One incidence of alopecia and crust on the right shoulder and peri-orbital staining was observed in a sham-operated animal during the 39th week post-implantation.

Both of these incidents resolved spontaneously by the following week.

### 3.2. Body Weight, Clinical Pathology and Urinalysis

The group mean body weight values and gains were largely similar between the two groups and showed the expected gradual increase over time. Results of urinalysis were also comparable between the two test groups, with no clear treatment-related effects observed. The values were within the normal expected range, as anticipated. Additionally, group mean hematology, biochemistry, and coagulation values were mostly comparable between the two test groups. While there were some statistically significant differences, these values were either within the normal range or below the normal range and did not have any biological significance.

### 3.3. Macroscopic Assessment of the Implantation Site

No alterations in normal structure or enlarged regional lymph nodes were observed in any of the animals at all time points.

At the 4-week time point, no local tissue reaction was recorded in both groups, and the expected location of the implant was confirmed in all animals from the balloon-implanted group.

For animals assigned to the 26-week time point, local tissue reactions included hematoma-like lesions around the operated area, noted in one sham-operated animal, encapsulation around the implant in all balloon-implanted animals, and the presence of adhesions around the operated area was noted in one sham-operated animal.

In animals assigned to the 52-week time point, local tissue reactions were observed, including a round whitish semi-solid mass of approximately 0.5 cm in diameter with a glandular appearance located laterally to the implantation site on the left side, noted in one balloon-implanted animal. Encapsulation around the implant was noted in three balloon-implanted animals, and the presence of adhesions around the operated area was noted in five balloon-implanted animals. The expected location of the implant was confirmed in most animals of the balloon-implanted group through various methods such as visualization of sutures, visualization of the balloon within the encapsulation, and palpation of the balloon.

### 3.4. Histopathological Evaluation

#### 3.4.1. 4-Week Time Point

In Group 1M (sham-operated control), histopathological analysis of the subcutaneous, perirectal area revealed minimal chronic inflammation, characterized by granulation tissue predominated by fibrosis with limited mixed mononuclear inflammatory cell infiltration (Figure 1).

In Group 2M (balloon-implanted), the presence of the implant was confirmed in 9 out of 10 animals. Histological examination of the subcutaneous, perirectal implantation site revealed a central cavity containing distinct elongated remnants of the balloon (Figure 2). Surrounding this cavity was relatively mature connective tissue classified as Grade 2. The inner surface of the capsule that enveloped the cavity was lined with a single layer of flattened to cuboidal macrophages, designated as Grade 1. Minimal degradation (Grade 1) of the balloon (implant) was observed, along with minor dissolution at the edges, cracks within the implant, and small fragments present. The calculated Reactivity Score, determined by subtracting the total score of the sham-operated control, fell within the category of “slight reaction” (score: 5) for the balloon (Table 5). No histopathological changes were detected in the kidneys, rectum, regional lymph nodes, or any adjacent tissues included in the sections of the implantation site.

#### 3.4.2. 26-Week Time Point

In the sham-operated control group, no histopathological changes were observed at this time point in the skin, underlying subcutis, or in the rectum (i.e., implantation site) (Figure 3). 

In the balloon-implanted group, the implantation site was identifiable in all 10 animals, and remnants of the balloon were still detectable in 6 out of 10 animals. Over time, the implants exhibited gradual fragmentation, resulting in the formation of multiple chambers (Figure 4). Each chamber was surrounded by a fibrotic reaction, primarily graded as minimal to mild (Grade 1 to 2), with occasional instances of moderate (Grade 3) fibrotic reactions. The central cavity housing the implant remnants was encased by relatively mature connective tissue, graded between 2 and 3. Minimal sporadic polymorphonuclear cells and lymphocytic infiltrations were sporadically observed within the capsule. A layer of flattened to columnar macrophages (graded 1 to 2) lined the inner surface of the capsule.

The observed changes were consistent with expectations for biodegradable materials and indicated a moderate to abundant degradation (Grade 2 to 4) of the balloon (implant). This ranged from the formation of some fragments, with either no or a moderate amount (Grade 2) of residual material, to abundant degradation with almost complete fragmentation (Grade 4). After subtracting the control data, the reactivity score indicated a “moderate reactivity” (score: 10.2) of the implantation site compared to the non-implanted site (Table 6). 

No histopathological changes were detected in the kidneys, rectum, regional lymph nodes, or any adjacent tissues included in the sections of the implantation site.

#### 3.4.3. 52-Week Time Point

In the sham-operated control group, there were no histopathological changes observed in the skin, underlying subcutis, or the rectum (i.e., implantation site) (Figure 5).

In the balloon-implanted animals, the implantation site was identifiable in 9 out of 10 animals. Notably, it appeared that the implant fragmented over time, resulting in the formation of multiple chambers (Figure 6). Each chamber exhibited a minimal fibrotic reaction, scored as Grade 1 (minimal). The entire implantation area was enveloped by relatively mature connective tissue, graded as Grade 2. Within the capsules, minimal sporadic lymphocytic infiltrations were noted. A layer of flattened to cuboidal macrophages (graded as Grade 1) lined the inner surfaces of the cavities where the implant had previously resided and was subsequently resorbed.

Upon calculating the Total Reactivity Score, following the subtraction of values from the sham-operated group, it indicated a “slight reactivity” (score: 8) of the balloon when compared to the non-implanted site (Table 7). 

## 4. Discussion

The principal findings of this study indicate that balloon implantation led to localized tissue responses characterized by fibrotic reactions and encapsulation, which are consistent with the anticipated foreign body reaction to biodegradable materials [25,26,27]. Notably, these responses were mild to moderate in nature, suggesting a gradual and expected healing process rather than adverse reactions.

The computation of the total Reactivity Score revealed a notable elevation in reactivity from the 4-week to the 26-week time interval, primarily attributed to heightened fibrotic responses. These responses are recognized as integral facets of the expected and progressive healing process, rather than adverse reactions, culminating in the development of scar tissue. Conversely, an observable decrease in the total Reactivity Score at the 52-week time point in comparison to the 26-week milestone suggests a diminishing inflammatory reaction. Significantly, this decline is construed as a manifestation of ongoing and anticipated healing, devoid of any untoward consequences.

The selection of an appropriate radiation dose for evaluating the impact on the integrity of the balloon implant remains a critical consideration in our study. While conventional prostate radiotherapy typically prescribes doses exceeding 70 Gy, it is essential to note that only a fraction of this dosage directly affects the rectum and the implant area. Thus, our choice to expose the implant to a dose of 10 Gy/5f was a cautious approach aimed at simulating a worst-case scenario specifically targeting the implant site. This decision was made to rigorously assess the potential degradation of the balloon under conditions that closely mimic its real-world application. Furthermore, comprehensive validation tests were conducted, including exposure of the balloon to doses as high as 150 Gy, demonstrating its robustness and resistance to degradation even at significantly elevated radiation levels. These findings reinforce the confidence in the balloon’s durability under realistic clinical radiation scenarios, supporting its potential feasibility for long-term use in prostate radiotherapy. Therefore, while acknowledging the discrepancy between our applied dose and the conventional prescription, our study provides valuable insights into the resilience of the implant to radiation exposure relevant to its intended clinical use. 

Our histopathological examinations meticulously tracked tissue reactions associated with the implantation of the biodegradable spacer and the effects of ionizing radiation. Importantly, our observations consistently revealed that all identified histological changes were exclusively attributed to the presence and degradation process of the spacer material. Notably, the administered ionizing radiation, within the parameters applied in our study, did not yield any discernible histopathological alterations or reactions in the examined tissues. These findings underscore the specificity of the histological reactions solely to the presence of the spacer and its degradation, reinforcing the notion that the ionizing radiation, as utilized in our experimental design, did not contribute to any observed tissue changes. 

These findings underscore the safety and biocompatibility of the balloon as a rectal spacer. The observed tissue responses align with the expected healing process, and the minimal degradation of the implant at the first weeks after implantation suggests its suitability for clinical use. Furthermore, the absence of adverse effects in adjacent tissues highlights the precision and efficacy of this spacer in minimizing radiation exposure to the rectum.

The results of this study have significant implications for the clinical use of balloon implants. The progressive and predictable degradation behavior of these implants underscores their potential utility in clinical settings, particularly in radiotherapy treatments. The absence of significant adverse effects on animal health and the observed tissue responses affirm the biocompatibility of the balloon implants. The findings presented here provide a strong foundation for further investigations and clinical trials. The use of these implants may offer improved patient outcomes and treatment precision while minimizing long-term implant-related complications.

## 5. Conclusions

Our study demonstrates the safety, biocompatibility, and gradual degradation behavior of the Bioprotect biodegradable fillable balloon as a rectal spacer in a rat model over a 52-week evaluation period. The observed tissue responses, characterized by mild to moderate fibrotic reactions and encapsulation, align with expected healing processes without adverse effects on adjacent tissues. These findings highlight the potential clinical utility of the Bioprotect balloon in minimizing radiation exposure to the rectum during prostate cancer radiotherapy. The predictable degradation pattern and minimal adverse reactions suggest its viability for clinical translation. Our research provides a foundation for future clinical trials, emphasizing improved patient outcomes and reduced long-term complications in prostate cancer treatment. 

## Figures and Tables

**Figure 1 pharmaceutics-15-02744-f001:**
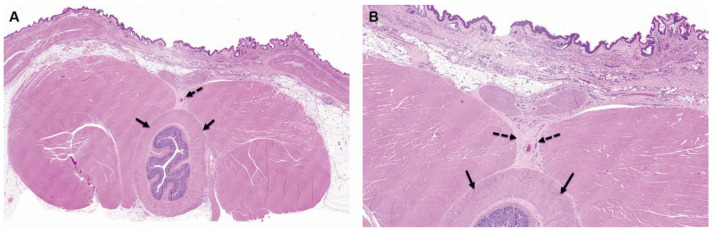
Group 1M (sham-operated control) at the 4-week time point, both low ×1 (**A**) and high ×4 (**B**) magnification views of the implantation site are presented. At this stage, changes are evident at the subcutaneous and perirectal implantation site, characterized by minimal chronic inflammation, which includes granulation tissue. This inflammation is primarily composed of fibrosis with minimal mononuclear inflammatory cells, as indicated by the broken arrows. However, no changes are observed in the rectum, as shown by the arrows in the images. The staining method used is Hematoxylin and Eosin (H&E).

**Figure 2 pharmaceutics-15-02744-f002:**
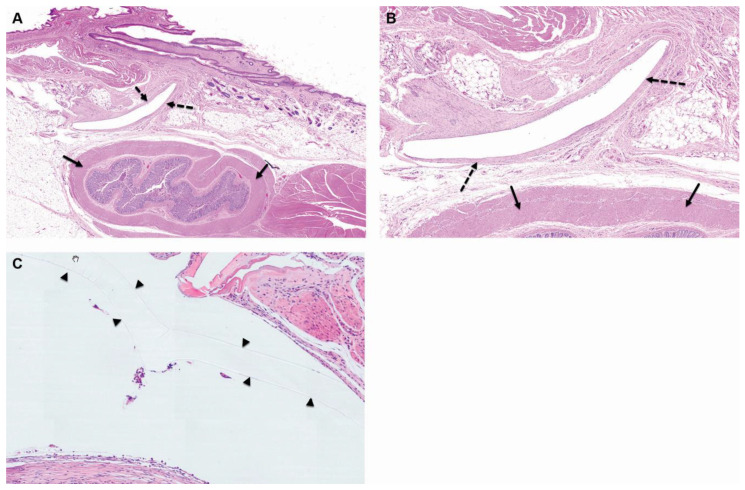
Group 2M (balloon-implanted group) at the 4-week time point, low ×1 (**A**), high ×4 (**B**), and very high (**C**) magnification views of the implantation site are presented. At this stage, changes are observed at the perirectal implantation site, which consist of a central cavity in which clear elongated remnants of the balloon are identified (arrowheads). The cavity is surrounded by relatively mature connective tissue, which has been scored as Grade 2, signifying mild changes (as represented by the broken arrows). Furthermore, a single layer of flattened to cuboidal macrophages is lining the inner surface of the capsule, and this has been scored as Grade 1, indicating minimal changes. Additionally, there are sporadic minimal polymorphonuclear cells and lymphocytes dispersed within the capsule. It is noteworthy that no changes are observed in the rectum, as indicated by the arrows. The staining method used for this image is Hematoxylin and Eosin (H&E).

**Figure 3 pharmaceutics-15-02744-f003:**
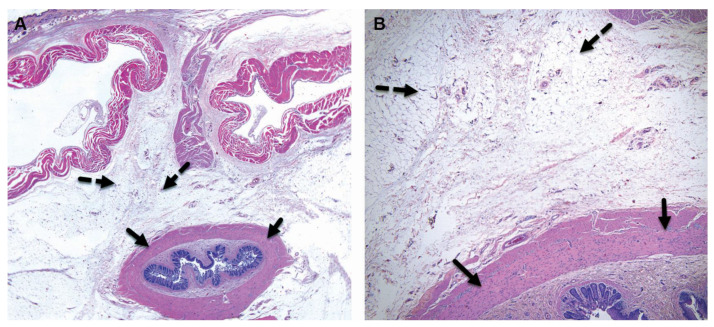
Group 1M (sham-operated control) at the 26-week time point, both low ×1 (**A**) and high ×4 (**B**) magnification views of the implantation site are presented. At this stage, there are no observable changes at the subcutaneous, perirectal implantation site, as indicated by the broken arrows. Additionally, there are no changes seen in the rectum, as indicated by the arrows. The staining method used for this image is Hematoxylin and Eosin (H&E).

**Figure 4 pharmaceutics-15-02744-f004:**
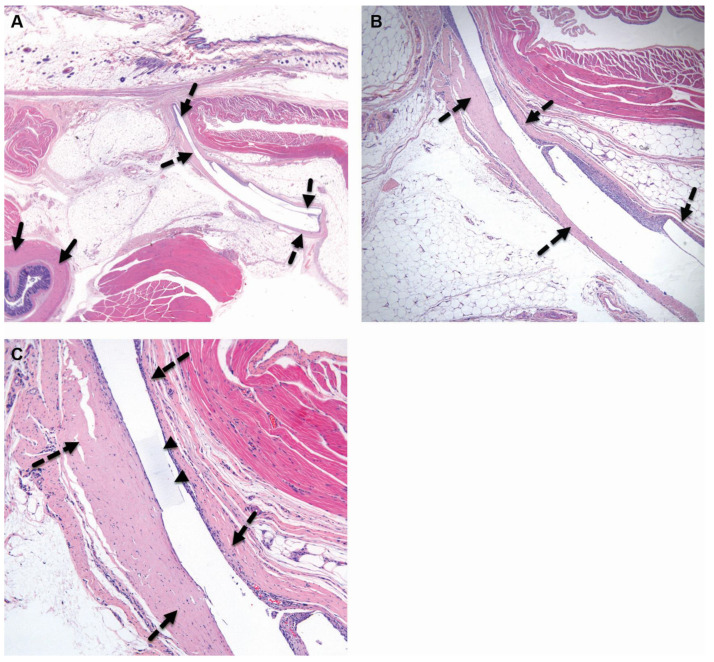
Group 2M (balloon-implanted group) at the 26-week time point, low ×1 (**A**), high ×4 (**B**), and very high (**C**) magnification views of the implantation site are presented. At this stage, changes are observed at the perirectal implantation site, which consist of a central cavity in which clear elongated remnants of the balloon are identified (arrowheads). It appears that the implant has slightly fragmented over time, forming multiple chambers, each surrounded by a fibrotic reaction, which is scored as grade 1. The cavity itself is surrounded by relatively mature connective tissue, scored as grade 2, mild (broken arrows). A single layer of flattened to cuboidal macrophages lines the inner surface of the capsule. Sporadic (minimal) polymorphonuclear cells and lymphocytes are dispersed within the capsule. No changes are observed in the rectum (arrows). H&E staining was used for this evaluation.

**Figure 5 pharmaceutics-15-02744-f005:**
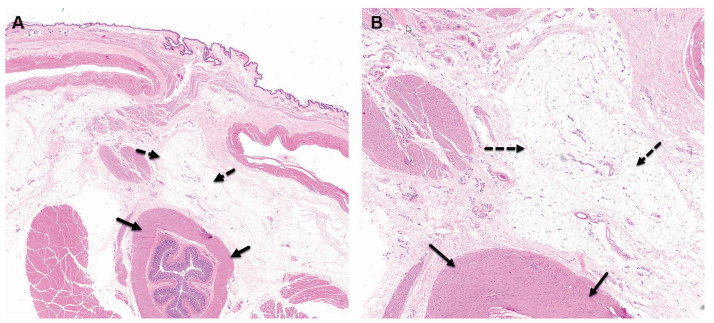
Group 1M (sham-operated control) at the 52-week time point, both low ×1 (**A**) and high ×4 (**B**) magnification views of the implantation site are presented. At this stage, no changes are observed at the subcutaneous, perirectal implantation site (broken arrows). No changes are seen in the rectum (arrows). H&E staining.

**Figure 6 pharmaceutics-15-02744-f006:**
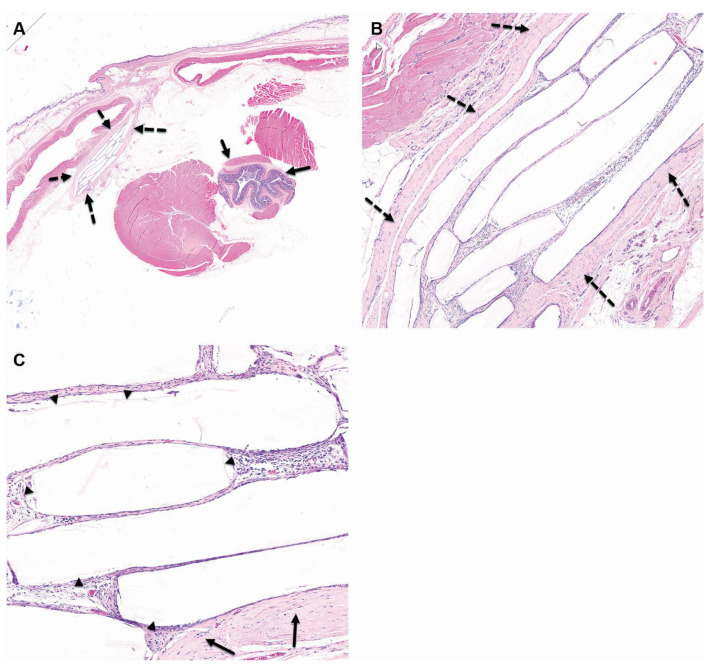
Group 2M (balloon-implanted group) at the 52-week time point, low ×1 (**A**), high ×4 (**B**), and very high (**C**) magnification views of the implantation site are presented. At this stage, histological alterations are observed at the perirectal implantation site, characterized by the presence of a central cavity containing ghost-like elongated remnants of the balloon (indicated by arrowheads). Over time, the implant appears to have fragmented, resulting in the formation of multiple chambers, each surrounded by a fibrotic reaction that is graded as mild (Grade 1). The central cavity is encircled by relatively mature connective tissue, with a mild intensity score of 2 (Grade 2) (indicated by broken arrows). Additionally, a monolayer of flattened to cuboidal macrophages lines the inner surface of these chambers. Scattered polymorphonuclear cells and lymphocytes are sporadically dispersed within the capsule but at minimal levels. Importantly, no discernible changes are observed in the rectal tissue (indicated by arrows). These findings were visualized using Hematoxylin and Eosin (H&E) staining.

**Table 1 pharmaceutics-15-02744-t001:** Study design.

Group No.	Group Size	Implantation & Iradiation			Scheduled Sacrifice (Post-Implantation)
		Implanted Material	Surgical Procedure	Irradiation Dose and Frequency	
1M	*n* = 10	Not Applicable (Sham-Operated Control)	Creation of perirectal SC “pocket” and placement of 2 sutures on both sides	Irradiation of 2 Gray/animal/day starting 7 days post-implantation × 5 days	4 Weeks
	*n* = 10				26 Weeks
	*n* = 10				52 Weeks
2M	*n* = 10	Balloon Spacer (Test Device)	Creation of perirectal SC “pocket” and implantation of 1 TD/animal and suturing the TD with 2 sutures		4 Weeks
	*n* = 10				26 Weeks
	*n* = 10				52 Weeks

SC = Subcutaneous; TD = Test Device.

**Table 2 pharmaceutics-15-02744-t002:** Histological evaluation system—tissue response.

Cell Type/Response	Score				
	0	1	2	3	4
Neovascularisation	0	Minimal capillary proliferation, focal, 1 to 3 buds	Groups of 4 to 7 capillaries with supporting fibroblastic structures	Broad band of capillaries with supporting fibroblastic structures	Extensive band of capillaries with supporting fibroblastic structures
Fibrosis	0	Narrow band	Moderately thick band	Thick band	Extensive band
Fatty Infiltrate	0	Minimal amount of fat associated with fibrosis	Several layers of fat and fibrosis	Elongated and broad accumulation of fat cells at activation site	Extensive fat completely surrounding the activation site

**Table 3 pharmaceutics-15-02744-t003:** Histological evaluation system—cell type/response.

Cell Type/Response	Score				
	0	1	2	3	4
Polymorphonuclear	0	Rare, 1–5/phf ^a^	5–10/phf	Heavy infiltrate	Packed
Lymphocytes	0				
Plasma Cells	0				
Macrophages	0				
Giant Cells	0	Rare, 1–2/phf	3–5/phf		Sheets

^a^—phf = per high powered (400×) field.

**Table 4 pharmaceutics-15-02744-t004:** Reactivity grade.

	Score Following Control Subtraction			
	0.0–2.9	3.0–8.9	9.0–15.0	>15.1
Total Reactivity	minimal or no reaction	slight reaction	moderate reaction	severe reaction

**Table 5 pharmaceutics-15-02744-t005:** Individual and group reaction scoring (4 weeks post-implantation).

		Histopathology Findings at 4 Weeks Post-Implantation (Group No., Treatment, Animal’s No.)																				
		1M											2M									
		Sham-Operated Control											Balloon Spacer (Test Device = TD)									
		1	2	3	4	5	6	7	8	9		10	31	32	33	34	35	36	37	38	39	40
Parameter		Sham-Operated Site											Implantation Site									
Cell Type/Response	Polymorphonuclear	0	0	0	0	0	0	0	0	0		0	1	1	1	1	1	1	1	1	1	NA
	Lymphocytes	1	1	1	1	1	1	1	1	1		1	1	1	1	1	1	1	1	1	1	NA
	Plasma Cells	0	0	0	0	0	0	0	0	0		0	0	0	0	0	0	0	0	0	0	NA
	Macrophages	0	0	0	0	0	0	0	0	0		0	1	1	1	1	1	1	1	1	1	NA
	Giant Cells	0	0	0	0	0	0	0	0	0		0	0	0	0	0	0	0	0	0	0	NA
	Necrosis	0	0	0	0	0	0	0	0	0		0	0	0	0	0	0	0	0	0	0	NA
	Sub Total	1	1	1	1	1	1	1	1	1		1	3	3	3	3	3	3	3	3	3	NA
	Sub Total (×2)	2	2	2	2	2	2	2	2	2		2	6	6	6	6	6	6	6	6	6	NA
Tissue Response	Neovascularization	1	1	1	1	1	1	1	1	1		1	1	1	1	1	1	1	1	1	1	NA
	Fibrosis/fibrous capsule	1	1	1	1	1	1	1	1	1		1	2	2	2	2	2	2	2	2	2	NA
	Fatty Infiltrate	0	0	0	0	0	0	0	0	0		0	0	0	0	0	0	0	0	0	0	NA
	Sub Total	2	2	2	2	2	2	2	2	2		2	3	3	3	3	3	3	3	3	3	NA
Total		4	4	4	4	4	4	4	4	4		4	9	9	9	9	9	9	9	9	9	NA
Group Total Reactivity		40											81									
Average Reactivity		4.0											9.0									
Average TD minus Average Control		5.0																				
Total Reactivity		Slight Reaction																				
Hemorrhage		0	0	0	0	0	0	0	0	0	0	0	0	0	0	0	0	0	0	0	0	NA
Granuloma		0	0	0	0	0	0	0	0	0	0	0	0	0	0	0	0	0	0	0	0	NA
foreign debris		0	0	0	0	0	0	0	0	0	0	0	0	0	0	0	0	0	0	0	0	NA
Tissue ingrowth		0	0	0	0	0	0	0	0	0	0	0	0	0	0	0	0	0	0	0	0	NA
Mineralization		0	0	0	0	0	0	0	0	0	0	0	0	0	0	0	0	0	0	0	0	NA
Residual Material (implant)		NA	NA	NA	NA	NA	NA	NA	NA	NA	NA	NA	1	1	1	1	1	1	1	1	1	NA

NA—not applicable (since the Test Device could not be detected/was not present in the sham control).

**Table 6 pharmaceutics-15-02744-t006:** Individual and group reaction scoring (26 weeks post-implantation).

		Histopathology Findings at 26 Weeks Post-Implantation (Group No., Treatment, Animal’s No.)																				
		1M											2M									
		Sham-Operated Control											Balloon Spacer (Test Device = TD)									
		11	13	14	15	16	17	18	19	20		61	41	42	43	44	45	46	47	48	49	50
Parameter		Sham-Operated Site											Implantation Site									
Cell Type/Response	Polymorphonuclear	0	0	0	0	0	0	0	0	0		0	0	1	1	1	1	3 *	0	1	0	1
	Lymphocytes	0	0	0	0	0	0	0	0	0		0	1	1	1	1	1	1	1	1	1	1
	Plasma Cells	0	0	0	0	0	0	0	0	0		0	0	0	0	0	0	0	0	0	0	0
	Macrophages	0	0	0	0	0	0	0	0	0		0	2	1	1	1	1	1	1	1	1	1
	Giant Cells	0	0	0	0	0	0	0	0	0		0	0	0	0	0	0	1	0	0	0	0
	Necrosis	0	0	0	0	0	0	0	0	0		0	0	0	0	0	0	0	0	0	0	0
	Sub Total	0	0	0	0	0	0	0	0	0		0	3	3	3	3	3	6	2	3	2	3
	Sub Total (×2)	0	0	0	0	0	0	0	0	0		0	6	6	6	6	6	12	4	6	4	6
Tissue Response	Neovascularization	0	0	0	0	0	0	0	0	0		0	1	1	1	1	1	1	1	1	1	1
	Fibrosis/fibrous capsule	0	0	0	0	0	0	0	0	0		0	2	2	2	3	3	3	2	2	2	2
	Fibrosis/fibrous within the cavity	0	0	0	0	0	0	0	0	0		0	3	1	1	2	1	2	0	2	2	1
	Fatty Infiltrate	0	0	0	0	0	0	0	0	0		0	0	0	0	0	0	0	0	0	0	0
	Sub Total	0	0	0	0	0	0	0	0	0		0	6	4	4	6	5	0	3	5	5	4
Total		0	0	0	0	0	0	0	0	0		0	12	10	10	12	11	18	7	11	9	10
Group Total Reactivity		0											92									
Average Reactivity		0.0											10.2									
Average TD minus Average Control		10.2																				
Total Reactivity		Moderate Reaction																				
Hemorrhage		0	0	0	0	0	0	0	0	0	0	0	0	0	0	0	0	0	0	0	0	0
Granuloma		0	0	0	0	0	0	0	0	0	0	1	1	0	0	0	0	0	0	0	0	0
foreign debris		0	0	0	0	0	0	0	0	0	0	1	1	0	0	0	0	2	0	0	0	0
Tissue ingrowth (fibrosis)		0	0	0	0	0	0	0	0	0	0	0	3	1	1	2	1	2	0	2	2	2
Mineralization		0	0	0	0	0	0	0	0	0	0	0	0	0	0	0	0	2	0	0	0	0
Residual Material (implant)		NA	NA	NA	NA	NA	NA	NA	NA	NA	NA	NA	4	2	2	3	4	2	4	3	4	2

NA—not applicable (since the Test Device was not present in the sham control). * Animal #46: The accumulation of polymorphonuclear cells was consistent with the morphological aspect of abscess formation, and was associated with the presence of bacterial colonies, mineralization, and hair shafts. The cavity was partially lined by squamous epithelium. These changes were suggested to occur due to incidental bacterial contamination at the implant site. Consequently, this case was excluded from the average calculation.

**Table 7 pharmaceutics-15-02744-t007:** Individual and group reaction scoring (52 weeks post-implantation).

		Histopathology Findings at 52 Weeks Post-Implantation (Group No., Treatment, Animal’s No.)																				
		1M											2M									
		Sham-Operated Control											Balloon Spacer (Test Device = TD)									
		21	22	23	24	25	26	27	28	29		30	51	52	53	101	55	56	57	58	59	60
Parameter		Sham-Operated Site											Implantation Site									
Cell Type/Response	Polymorphonuclear	0	0	0	0	0	0	0	0	0		0	0	0	0	0	NA	0	0	0	0	0
	Lymphocytes	0	0	0	0	0	0	0	0	0		0	1	1	1	1	NA	1	1	1	1	1
	Plasma Cells	0	0	0	0	0	0	0	0	0		0	0	0	0	0	NA	0	0	0	0	0
	Macrophages	0	0	0	0	0	0	0	0	0		0	1	1	1	1	NA	1	1	1	1	1
	Giant Cells	0	0	0	0	0	0	0	0	0		0	0	0	0	0	NA	0	0	0	0	0
	Necrosis	0	0	0	0	0	0	0	0	0		0	0	0	0	0	NA	0	0	0	0	0
	Sub Total	0	0	0	0	0	0	0	0	0		0	2	2	2	2	NA	2	2	2	2	2
	Sub Total (×2)	0	0	0	0	0	0	0	0	0		0	4	4	4	4	NA	4	4	4	4	4
Tissue Response	Neovascularization	0	0	0	0	0	0	0	0	0		0	1	1	1	1	NA	1	1	1	1	1
	Fibrosis/fibrous capsule	0	0	0	0	0	0	0	0	0		0	2	2	2	2	NA	2	2	2	2	2
	Fibrosis/fibrous within the cavity	0	0	0	0	0	0	0	0	0		0	1	1	1	1	NA	1	1	1	1	1
	Fatty Infiltrate	0	0	0	0	0	0	0	0	0		0	0	0	0	0	NA	0	0	0	0	0
	Sub Total	0	0	0	0	0	0	0	0	0		0	4	4	4	4	NA	4	4	4	4	4
Total		0	0	0	0	0	0	0	0	0		0	8	8	8	8	NA	8	8	8	8	8
Group Total Reactivity		0											72									
Average Reactivity		0.0											8.0									
Average TD minus Average Control		8.0																				
Total Reactivity		Slight Reaction																				
Hemorrhage		0	0	0	0	0	0	0	0	0	0	0	0	0	0	0	NA	0	0	0	0	0
Granuloma		0	0	0	0	0	0	0	0	0	0	0	0	0	0	1	NA	0	0	0	0	1
foreign debris		0	0	0	0	0	0	0	0	0	0	0	0	0	0	0	NA	0	0	0	0	0
Tissue ingrowth (fibrosis)		0	0	0	0	0	0	0	0	0	0	0	2	2	2	2	NA	2	2	2	2	2
Mineralization		0	0	0	0	0	0	0	0	0	0	0	0	0	0	0	NA	0	0	0	0	0
Residual Material (implant)		NA	NA	NA	NA	NA	NA	NA	NA	NA	NA	NA	4 *	4 *	4 *	4 *	NA	4 *	4 *	4 *	4 *	4 *

* Minimal ghost residues, judged to be innocuous, reflecting end-of-process residue in trace amounts.

## Data Availability

The data presented in this study are available on request from Shaul Shohat.

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
