# Peer review of "Assessment of Bioprotect’s Biodegradable Balloon System as a Rectal Spacer in Radiotherapy: An Animal Study on Tissue Response and Biocompatibility"

_pharmaceutics, 2023, doi:10.3390/pharmaceutics15122744_

Round 1
Reviewer 1 Report
Comments and Suggestions for Authors
1. Table 1---The radiation dose in this study was 10Gy/5f, while the conventional prostate radiotherapy dose was 70Gy or above. At higher radiation doses, whether the degradation process of the balloon would change should be considered.
2. Line 261 What data should be indicated for statistical analysis? The value of p provided or not?
3. Line 328--- The shape change of the implant may cause the deformation of the surrounding tissue, which will affect the treatment accuracy. Is the shape change of the balloon drastic in the first few weeks after implantation?
4. Table 6--
please check the calculation again.
Comments on the Quality of English Language
No
Author Response
Reviewer 1
- Table 1---The radiation dose in this study was 10Gy/5f, while the conventional prostate radiotherapy dose was 70Gy or above. At higher radiation doses, whether the degradation process of the balloon would change should be considered.
Reply: Thank you for your insightful comments and your attention to the radiation dosage in our study as outlined in Table 1.
Regarding your first point about the conventional prostate radiotherapy dose exceeding 70Gy, we completely understand the discrepancy between our applied dose of 10Gy/5f and the typical prescribed dose. However, our rationale for utilizing this lower dose was based on the consideration that while the prescribed dose for prostate radiation indeed surpasses 70Gy, only a fraction of this dosage directly impacts the rectum and the implant area. Our choice of 10Gy aimed to simulate a worst-case scenario that directly affects the implant site, ensuring a cautious evaluation of balloon integrity under these circumstances.
Furthermore, we acknowledge your interest in understanding the balloon's performance under higher radiation doses. We have rigorously validated the durability of the balloon by subjecting it to an extensive validation protocol, including exposure to 150Gy. The comprehensive testing confirmed the balloon's robustness, showing no discernible deterioration even at significantly higher radiation levels.
We have incorporated a detailed explanation of our rationale for the selected radiation dose and the balloon's resilience to higher doses in the Discussion section of the manuscript.
- Line 261 What data should be indicated for statistical analysis? The value of p provided or not?
Reply: As explained in our Materials and Methods section, thorough statistical analysis was conducted in our study. Some hematology and blood biochemistry parameters showed statistical significance (either p<0.05 or p<0.01). However, these values fell within or below the normal range, lacking biological significance. As a result, we did not specifically mention these values in the text. The relevant information is included in the results section of the manuscript.
- Line 328--- The shape change of the implant may cause the deformation of the surrounding tissue, which will affect the treatment accuracy. Is the shape change of the balloon drastic in the first few weeks after implantation?
Reply: Thank you for your insightful comment regarding the potential impact of implant shape change on surrounding tissue and its implications for treatment accuracy.
In response to your query, we want to highlight that the radiation treatment plan is meticulously crafted based on the patient's anatomy acquired from a CT scan conducted approximately one week post balloon implantation. This timeframe allows for the stabilization of the anatomical structures, ensuring that the treatment plan is designed on the foundation of a relatively stable formation of the patient's anatomy.
Furthermore, it is essential to emphasize that the treatment plan involves vigilant monitoring of patients who undergo the implant procedure. These individuals undergo regular CT scans on a weekly basis as part of the protocol. To date, our observations from these follow-up scans have not revealed any discernible shape changes or deformations caused by the presence of the balloon implant.
The consistency and stability of the anatomical structures, as evidenced by the absence of observed shape alterations in the patients undergoing regular monitoring, provide reassurance regarding the reliability and accuracy of the treatment plan based on the initial CT scan post-implantation.
- Table 6-- please check the calculation again.
Reply: We appreciate the reviewer's scrutiny of our calculations, which were indeed found to be inaccurate. The discrepancy stemmed from animal 46. In this case, there was an accumulation of polymorphonuclear cells consistent with an abscess formation, along with bacterial colonies, mineralization, and hair shafts. The cavity had partial coverage by squamous epithelium. These changes were likely caused by an accidental bacterial presence at the implant site. As a result, we excluded this case from the average calculation and added a footnote to the table to clarify this observation.
Reviewer 2 Report
Comments and Suggestions for Authors
Dear colleagues,
In this manuscript, the authors suggest bioprotect's biodegradable balloon system as a rectal spacer in radiotherapy. The study is performed on experimental material with sufficient amount of animals in each group. The results are interesting. The figures reflect the results of the study. Despite the very good impression of the article, there are some questions which could improve the article in my opinion, partly:
Authors didn’t use the results of their statistical calculation in the main part, in the discussion which should be accent effects of their manuscript. It could be also significant improvement if authors compare their results with other work in the discussion.
Morphometric data of histological changes are desirable.
Conclusions are not presented in that version of the article, but it could be important for readers, especially with using digital results there. That should be included in the summary also.
The article needs stylistic correction as many parts are presented as notes of an experiment but not as article text.
In summary, I have been satisfied with the high level of the article. I believe this manuscript will attract significant attention from the research community. In my personal opinion, the article is very valuable, a great prospect for further research, and, after corrections, can be recommended for publication in Pharmaceutics.
Author Response
Reviewer 2
Dear colleagues,
In this manuscript, the authors suggest bioprotect's biodegradable balloon system as a rectal spacer in radiotherapy. The study is performed on experimental material with sufficient amount of animals in each group. The results are interesting. The figures reflect the results of the study. Despite the very good impression of the article, there are some questions which could improve the article in my opinion, partly: Morphometric data of histological changes are desirable.
Reply: We evaluated the histological parameters using a detailed scoring system, as outlined in the manuscript. Our study did not include morphometric evaluations, which are typically not conducted in similar studies
Authors didn’t use the results of their statistical calculation in the main part, in the discussion which should be accent effects of their manuscript. It could be also significant improvement if authors compare their results with other work in the discussion.
Reply: As explained in our Materials and Methods section, thorough statistical analysis was conducted in our study. Some hematology and blood biochemistry parameters showed statistical significance (either p<0.05 or p<0.01). However, these values fell within or below the normal range, lacking biological significance. As a result, we did not specifically mention these values in the text. The relevant information is included in the results section of the manuscript.
We appreciate your insightful recommendation and acknowledge the potential value of comparing our findings with existing studies in the field. However, we wish to highlight that to the best of our knowledge, there is a scarcity of comprehensive clinical and histopathological evaluations specifically focusing on rats undergoing implantation of a biodegradable rectal spacer, akin to the detailed assessments conducted in our study.
Regrettably, due to the absence of similar studies that closely match our experimental design and evaluations, we encountered limitations in directly comparing our results with previously published research. As a result, we could not provide comparative data from prior works in our discussion section.
It is important to note that our study is intended to fill this gap by offering a meticulous evaluation framework and in-depth analysis of the effects of the biodegradable rectal spacer in rat models. Consequently, we have emphasized in our manuscript that our results contribute novel insights to the field and hold the potential to serve as a benchmark for future studies seeking to explore similar interventions in preclinical settings.
Conclusions are not presented in that version of the article, but it could be important for readers, especially with using digital results there. That should be included in the summary also.
Reply: We've now included a conclusions paragraph at the end of the article and also added a concluding sentence to the abstract
The article needs stylistic correction as many parts are presented as notes of an experiment but not as article text.
Reply: We appreciate the reviewer's feedback. In response, we've revised significant portions of the text, particularly in the Materials and Methods section, aiming to make the writing style more aligned with a manuscript and less akin to an experimental report, as suggested.
In summary, I have been satisfied with the high level of the article. I believe this manuscript will attract significant attention from the research community. In my personal opinion, the article is very valuable, a great prospect for further research, and, after corrections, can be recommended for publication in Pharmaceutics.
Reviewer 3 Report
Comments and Suggestions for Authors
General point. “The objective of this investigation was to comprehensively evaluate the localized tissue response subsequent to the implantation of a perirectally positioned balloon in a rat model followed by a course of irradiation consisting of five consecutive days to mimic the clinical scenario expected in humans.“ There is one important difference with the clinical scenario, namely the absence of a cancer. A second putative concern is the low energy of the X-rays. This will certainly result in a dose distribution that is largely different from the clinical one. No isodoses are shown in the present study. This aspect of the experimental set up should be discussed.
Comments:
One. The semi-quantitative histological scoring, tabulation and statistical evaluation are exemplary.
Two. Figures need scale bars and windows for higher power fields. More extensive labeling of organs and tissues would facilitate the legibility of the histological pictures for the reader not trained in microscopic morphology. Indication of the animal number would permit comparison between Figures and Tables.
Three. Since unirradiated controls are missing the Discussion section should address the participation at the histological reaction due to ionizing radiation with that due to the implantation of the spacer.
Author Response
Reviewer 3
General point. “The objective of this investigation was to comprehensively evaluate the localized tissue response subsequent to the implantation of a perirectally positioned balloon in a rat model followed by a course of irradiation consisting of five consecutive days to mimic the clinical scenario expected in humans.“
There is one important difference with the clinical scenario, namely the absence of a cancer. A second putative concern is the low energy of the X-rays. This will certainly result in a dose distribution that is largely different from the clinical one. No isodoses are shown in the present study. This aspect of the experimental set up should be discussed.
Reply: We completely understand the discrepancy between our applied dose of 10Gy/5f and the typical prescribed dose. However, our rationale for utilizing this lower dose was based on the consideration that while the prescribed dose for prostate radiation indeed surpasses 70Gy, only a fraction of this dosage directly impacts the rectum and the implant area. Our choice of 10Gy aimed to simulate a worst-case scenario that directly affects the implant site, ensuring a cautious evaluation of balloon integrity under these circumstances.
Furthermore, we acknowledge your interest in understanding the balloon's performance under higher radiation doses. We have rigorously validated the durability of the balloon by subjecting it to an extensive validation protocol, including exposure to 150Gy. The comprehensive testing confirmed the balloon's robustness, showing no discernible deterioration even at significantly higher radiation levels.
We have incorporated a detailed explanation of our rationale for the selected radiation dose and the balloon's resilience to higher doses in the Discussion section of the manuscript.
Comments:
One. The semi-quantitative histological scoring, tabulation and statistical evaluation are exemplary.
Reply: We greatly appreciate your positive feedback.
Two. Figures need scale bars and windows for higher power fields. More extensive labeling of organs and tissues would facilitate the legibility of the histological pictures for the reader not trained in microscopic morphology. Indication of the animal number would permit comparison between Figures and Tables.
Reply: We've included arrows in each photo to highlight the main relevant organs and tissues. These are explained in the respective photo captions. Due to technical limitations, adding scale bars at this stage isn't feasible. We suggest retaining the explanation provided in the captions for magnification levels (low and/or high). Each caption clearly identifies whether the photo is from Group 1M (Sham Operated Control) or Group 2M (balloon-implanted) for easy reference.
Three. Since unirradiated controls are missing the Discussion section should address the participation at the histological reaction due to ionizing radiation with that due to the implantation of the spacer.
Reply: Our comprehensive histopathological evaluations have meticulously tracked and attributed all observed changes solely to the presence of the spacer and its associated degradation process. Importantly, it is noteworthy to highlight that the ionizing radiation administered in our experimental setup did not elicit any discernible histopathological changes.
We have ensured that our Discussion section explicitly addresses this crucial aspect, emphasizing that all identified histopathological changes were exclusively linked to the presence and degradation of the spacer, while affirming the absence of any discernible histological reactions attributable to the ionizing radiation employed.
Round 2
Reviewer 1 Report
Comments and Suggestions for Authors
Accept
Comments on the Quality of English LanguageNo
Reviewer 2 Report
Comments and Suggestions for Authors
Dear colleagues,
Despite my conclusion about the possibility of publishing an article in its present form, my opinion is that authors can significantly improve the article with wider discussion, morphometric data of histological slides, and escaping “experimental notes” stile in parts of the manuscript.